# Mental Health in the Post-Lockdown Scenario: A Scientometric Investigation of the Main Thematic Trends of Research

**DOI:** 10.3390/ijerph20136310

**Published:** 2023-07-06

**Authors:** Ilaria Cataldo, Dora Novotny, Alessandro Carollo, Gianluca Esposito

**Affiliations:** Department of Psychology and Cognitive Science, University of Trento, 38068 Rovereto, Italy

**Keywords:** mental health, psychological well-being, depression, anxiety, post-lockdown, COVID-19, document co-citation analysis, bibliometrics, visualized research

## Abstract

Since the outbreak of COVID-19, researchers and clinicians have published scientific articles on the SARS-CoV-2 virus and its medical, organizational, financial, and psychological implications. However, many effects have been observed in the post-lockdown scenario. In this study, we adopted a scientometric–bibliometric approach to drawing the state of the art regarding the emotional and psychological effects of the pandemic after the lockdown. In Scopus, we found 791 papers that were subsequently analyzed using CiteSpace. The document co-citation analysis (DCA) computation generated a network of eight major clusters, each representing a central area of investigation. Specifically, one major cluster—cluster no. 1—focuses on the long-term effects of the COVID-19 pandemic and individuals’ ability to develop adaptive coping mechanisms and resilience. The results allow us to frame the fields covered by researchers more precisely and the areas that still need more investigation.

## 1. Introduction

At the turn of 2019 and 2020, the world witnessed the COVID-19 pandemic, which is caused by SARS-CoV-2; this led to a multitude of consequences, starting from immediate closures to drastic changes in our everyday habits [1]. The term “*lockdown*” has been used extensively in reference to measures, such as confining the population of an entire country to their residential homes and blocking most activities and transportation in an effort to contain the spread of COVID-19 [2]. The need to adapt one’s daily routine to the imposed in-home solution, coupled with the fear of contagion and the potential loss of loved ones, drastically affected people’s mental health; symptoms that are the most associated with this impact include depression, anxiety, and distress [3,4,5,6]. Similarly, “*post-lockdown*” refers to the period following a lockdown. In the case of COVID-19, it is hard to outline a common time frame, due to the distinct measures adopted in different countries and at different times. As such, it is common to use post-lockdown as a term related to the period following each outbreak wave.

The alternating periods of in-home restrictions and periods with fewer restrictions, combined with a diffuse perception of uncertainty regarding people’s safety and the possibility of returning to a predictable routine, resulted in increased psychological issues [7,8] or, conversely, strengthened resilience, which can be described as the ability to recover from or cope with critical situations [9]. During the first period of the lockdown, Rossi et al. [10] observed a deterioration in mental health in the general Italian population, in terms of post-traumatic stress symptoms, depression, anxiety, insomnia, perceived stress, and adjustment disorder symptoms. However, contradictory results emerged from similar studies that investigated mental health during the lockdown [11]. For instance, Koenig et al. [12] observed no immediate effects on adolescents’ well-being following the initial closure of schools in Germany. Similarly, when using a machine learning approach to investigate the impact of time spent in the lockdown on individuals’ physical well-being, Carollo et al. [13] found that self-perceived loneliness decreased during the first weeks of the lockdown. This result was initially observed in participants living in the United Kingdom and replicated in a sample of participants living in Greece.

To better understand the effects of the COVID-19 pandemic in the post-lockdown periods on people’s mental health, the current work aims to identify the most impactful publications and the main thematic developments of research within the topic. To do so, we analyzed the available scientific literature using a scientometric–bibliometric approach. Scientometrics is a field that involves the integration of scientific mapping (i.e., visualization of the evolution of a research domain in time) and bibliometric analysis (i.e., application of quantitative techniques to bibliometric data) [14,15,16,17]. Compared to other review methods (e.g., narrative review, systematic review, meta-analysis), scientometrics allows identifying the most impactful documents and main thematic domains of research using a data-driven approach applied to examine the existing quantitative relationships between large samples of scientific publications. For these reasons, scientometrics allows reducing the risk of biases when assessing published research in a specific field [18,19]. In the present study, we opted for a document co-citation analysis (DCA) to produce a network of thematic clusters representing the main discussed topics related to mental health in the post-lockdown period. Each cluster, which will be qualitatively developed in detail, includes publications related to specific issues, with the possibility to easily track influential articles.

## 2. Materials and Methods

This current study follows the pipeline of recent publications in the field of scientometrics (e.g., refs. [20,21,22]). In Scopus, we conducted a literature search on 31st December 2022 using the following string: [TITLE-ABS-KEY((“post lockdown” OR “post lock-down” OR “after lockdown” OR “after lock-down” OR “post pandemic” OR “post-pandemic” OR “following lockdown” OR “following lock-down” OR “following pandemic”) AND (“COVID*” OR “coronavirus” OR “SARS-COV-2”) AND (“depress*” OR “anxi*” OR “stress*” OR “trauma*” OR “mental health” OR “psychiatr*”))]. The terms included in the string reflected the psychological symptoms of the general population after the COVID-19 lockdown. The search yielded 791 documents, which were published or accepted for publication between 2019 and 2023, and included references to 35,816 other documents. We opted to use Scopus over comparable archives (e.g., Web of Science, Medline) due to its broader coverage of recently published documents and indexed journals [23,24]. The downloaded documents were initially analyzed using the *bibliometrix* package for R developed by Aria and Cuccurullo [25] to uncover the most influential authors, frequent keywords, the most frequent countries appearing in affiliation strings, and the main sources for documents on mental health in the post-lockdown scenario.

### 2.1. Data Import on CiteSpace

Before proceeding with the scientometric analysis, the file containing 791 publications and their 35,816 references was imported into CiteSpace (version 6.1.R2, 64 bits), according to the suggested guidelines [26,27]. In this phase, all references are screened and regarded as valid only if all of the following pieces of information are available: title, year of publication, author(s), source, volume, pages, DOI [28]. From this initial screening, a total of 34,721 references (96.94% of the total) were recognized as valid. Data loss corresponded to 3.06% of the dataset, which can be considered inconsequential, as it falls below the acceptable range of 1.00–5.00% [29].

### 2.2. Document Co-Citation Analysis (DCA)

To identify the most impactful publications and main thematic domains of research, a DCA was then computed. DCA is a type of bibliometric analysis that focuses on the patterns of co-citations among documents (i.e., occurrences in which two or more documents are cited together by other publications) [30]. The general assumption behind DCA is that greater co-citation frequencies shared by documents indicate commonalities in terms of research interests and domains of research [31]. To model the patterns of co-citations among documents, DCA creates a network in which documents are included as single nodes and co-citations as edges. In the generated network, the frequencies of co-citations are included as edge weights [32]. Based on the network’s properties, groups of nodes reflecting similar research topics are found and clustered using CiteSpace’s clustering functions.

CiteSpace provides a set of node selection criteria for creating a balanced DCA network that captures the information embedded in the dataset. These criteria are the G-index, top *N*, and top *N*%. The G-index is an adaptation of the more popular H-index [33], with the difference that the G-index corresponds to the higher number equivalent to the average number of citations of the author’s most cited *g* number of publications [34,35]. In CiteSpace, G-index goes along with a scaling factor, *k*, to change the number of nodes included in the final output [29], where greater *k* values will generate DCA networks with more nodes. Lastly, top N and top *N*% have a similar function: the most cited *N* number or *N*% of references is selected within a time frame (from here on referred to as time slice) to be nodes. To maximize the amount of retrieved information, the time slice was kept at the value of 1 year for the current work. To determine the best DCA network, the results obtained by using specific selection criteria were compared. DCAs were generated with the following node selection criteria: G-index (*k* = 15, 25, 50), top *N* (*k* = 25, 50, 75), and top *N*% (*k* = 5, 10, 15). The network’s structural metrics (explained below), the number of nodes, links, and thematic clusters of research were used to compare the obtained DCAs and to choose the network with the optimal properties.

The steps from the identification of records to the included nodes are presented in Figure 1.

Once the optimal network was computed, CiteSpace’s “Find clusters” function was used to divide the network into separate thematic domains of research. Clusters’ labels were automatically generated with CiteSpace’s log-likelihood ratio (LLR) algorithm. The LLR algorithm creates a denomination through the identification of unique terms and expressions included in the titles of the contributing papers [26]. The LLR method provides the most accurate labels when compared to other automatic approaches available in CiteSpace. However, LLR labels, in some cases, might lack accuracy when compared to manually generated labels [36]. For this reason, both LLR and manually generated cluster labels will be provided in the manuscript.

### 2.3. DCA Network Evaluation Metrics

Many criteria are utilized to assess the created DCA networks. They are categorized as structural or temporal measures. Modularity Q, silhouette, and betweenness centrality are examples of structural metrics. Modularity is a measure of the overall network and its ability to be divided into discrete clusters [37]. The greater its value (ranging from 0 to 1), the more divided or different the network clusters are [31]. Moreover, the silhouette measures the homogeneity of each cluster (i.e., every cluster has its own silhouette measure), where greater estimates (from a minimum of −1 to a maximum of 1) denote a more significant homogeneity of the cluster [38,39]. Furthermore, betweenness centrality quantifies the goodness of the connection between a single node and two other nodes within the network [26,40]. Greater values depict a better connection of the node/publication within the network [39]. Other temporal metrics included in the analyses are citation burstiness and sigma, which evaluate the prominence of each node in the network. Kleinberg’s algorithm was adopted to calculate the citation burstiness [41]. In this computation, greater numbers (with a minimum value of 0) correspond to increased citations of the node (i.e., the publication) in a targeted period (i.e., time slice). As such, a high citation burstiness value depicts publications that have gained significant attention from the scientific community [42]. Sigma is computed through citation burstiness and betweenness centrality values following the formula ${\left(centrality+1\right)}^{burstiness}$, where greater scores indicate an elevated impact of the node over the network and, subsequently, suggest the novelty and significance of the publication.

## 3. Results

### 3.1. Bibliometric Analysis on the Citing References

The initial bibliometric analysis showed that each citing document obtained an average of 10.28 citations, with a yearly mean of 3.312 citations per document. The documents cited more frequently were authored by Singh et al. [43] (total citations = 644; total citations/year = 161) and Fancourt et al. [44] (total citations = 431; total citations/year = 143.7).

Collectively, 2033 keywords were selected by the authors in the included documents, with the most popular being “COVID-19” (*n* = 445 appearances), “mental health” (*n* = 120 appearances), “pandemic” (*n* = 99 appearances), “lockdown” (*n* = 81 appearances), “anxiety” (*n* = 80 appearances), “depression” (*n* = 77 appearances), “coronavirus” (*n* = 53 appearances), “COVID-19 pandemic” (*n* = 47 appearances), “stress” (*n* = 40 appearances), and “SARS-CoV-2” (*n* = 23 appearances). Figure 2 includes a graphical representation of the main patterns of co-occurrence between the keywords in the dataset.

In the data sample, 4719 distinct authors were identified, with 53 scholars having published 54 single-authored documents. On average, the network included 0.168 documents per author and 6.32 co-authors for each document. The three most productive authors in the data sample were Li J., Liu J., Wang Y., and Wang Y., respectively, with six, five, five, and five published documents.

The corresponding authors were more frequently affiliated with institutions from the United States of America (*N* = 100; frequency = 0.1464; single-country publications (SCP) = 79; multi-country publications (MCP) = 21), United Kingdom (*N* = 85; frequency = 0.1245; SCP = 66; MCP = 19), or Italy (*N* = 77; frequency = 0.1127; SCP = 57; MCP = 20). The graphical results are reported in Figure 3.

Ultimately, the main sources of documents emerged to be the *International Journal of Environmental Research and Public Health* (66 documents), *Frontiers in Psychiatry* (26 documents), and *Frontiers in Psychology* (24 documents).

### 3.2. Document Co-Citation Analysis

Following the parameter optimization procedure described in Section 2.2, the final DCA network was created by adopting top N = 25. The DCA network consisted of a total of 1896 nodes and 5776 links (i.e., approximately 3.05 links per node; see Figure 4). Looking at the structural metrics of the network, modularity Q was 0.9447, and the average silhouette was 0.9517. Based on these values, it is possible to infer that the network is highly divisible into highly coherent clusters.

Eight major thematic clusters were identified in the network. All eight clusters had their mean publication year in 2020. The largest cluster was cluster no. 1 (mean publication year = 2020), with a size of 86 documents. The most homogeneous clusters were no. 4, no. 11, no. 18, no. 43, and no. 47, all with a maximum silhouette score of (=1.00). This score is probably due to the fact that in all of these clusters, there is only one citing article. Details of the distinct clusters are reported in Table 1. Table 1 presents both the labels computed with the LLR method (LLR Label) and the ones created manually (suggested label).

Proceeding with the results, 43 documents displayed a significant citation burstiness within the network; however, 4 of these turned out to be repetitions of other documents already within this list (i.e., refs. [45,46,47,48]). For this reason, the final number of documents with significant burstiness strength was 39. In Table 2, the top 20 documents are presented, which are the most relevant to the discussion. The three articles showing the greatest values of citation burstiness are as follows: Brooks et al. [45] (citation burstiness = 29.18), which is a review debating the psychological impact of the lockdown; Xiong et al. [49] (citation burstiness = 15.99), which is a systematic review that synthesizes the literature on the psychological outcomes of COVID-19 on the general population and the associated risk factors; Holmes et al. [46] (citation burstiness = 8.43), which is an exploration of the psychological, social, and neuropsychological effects of COVID-19, discussing the most prominent priorities and identifying long-term strategies for scientific research in mental health.

## 4. Discussion

The purpose of this report is to review the research on the consequences of the COVID-19 epidemic on mental health following the lockdown. A DCA network with eight clusters was created using a scientometric approach. Furthermore, 39 documents with substantial citation bursts are identified.

Each cluster will be presented qualitatively, from the largest to the smallest, according to the cluster size (i.e., the number of documents included in the cluster). The citing papers included in each cluster are distinguished by their coverage (amount of references in that cluster reported in that paper) and overall citing score (GCS, meaning the total amount of citations retrieved by Scopus for that paper). The labels are named using a manual approach as it provides a more accurate representation of the clusters’ research themes [21].

### 4.1. Cluster No. 1: Long-Term Effects and Resilience

The main citing articles for cluster no. 1 (LLR title = ”Resilience”; silhouette = 0.853) are reported in Table 3. Within the macro-topic concerning the long-term effects of COVID-19 and resilience, the documents included in this cluster present a variety of arguments that can be grouped into sub-topics. As such, the documents will be discussed in narrower sections to facilitate the comprehension of the state of the art.

#### 4.1.1. Effects on People with Pre-Existing Low Health Conditions

##### Mood Disorders

An investigation of people diagnosed with bipolar disorder found differences in depression, anxiety, and somatic symptoms during and after the first wave in Austria (April–May 2020) [78]. People diagnosed with bipolar disorder experienced higher levels of psychological burdens compared to healthy controls in relation to the imposed social distancing. From the same study, it also emerged that people with bipolar disorder experienced greater bodily distress during the in-home period when compared to healthy individuals. However, it was observed that these somatization levels decreased after the end of the restrictions.

Regarding adults with major depressive disorder, Leightley et al. [72] investigated the trajectories of symptoms during the pre-, during, and post-lockdown periods. They found that sleep duration decreased during and after domestic isolation. Furthermore, the authors observed a slight decrease in symptoms and self-esteem; however, the findings did not reach statistical significance. Similarly, Siddi et al. [79] observed that the symptoms related to mood disorders were stable before, during, and after the lockdown, and that online social interactions increased, working as coping mechanisms to contrast loneliness. These results find agreement with the existing literature. For instance, by studying a convenience sample from the United Kingdom, Sommerlad et al. [116] observed that people with higher quality or more face-to-face or phone/video contacts reported fewer depressive symptoms. This association was stronger when considering contact quality as compared to contact quantity.

##### Anxiety

Min et al. [66] explored the long-term consequences of the pandemic aftermath, comparing the prevalence of mental issues in the general population before and after the lockdown in South Korea. They found that while suicidal rates had decreased, there were higher levels of depression, anxiety, and general distress, indicating maladaptive responses to the so-called “new normality” derived from prolonged social restrictions. The authors suggest that psychological distress could have also been elicited by the need to adapt to the “Fourth Industrial Revolution” accelerated by the pandemic, which led to the implementation of electronic technologies and online communications for everyone. This drastic change could have caused a form of stress induced by technological demands, resulting in physiological alterations caused by increased screen exposure and adaptation to a new routine [66].

An investigation performed during the first months of the lockdown in Indonesia highlighted that 20% of the respondents suffered from clinical anxiety. Young females were the most vulnerable to this risk, along with individuals suspected of having COVID-19, those with lower levels of education, and those with low education or weak social support [73]. As a limitation of this study, the authors mentioned that the participants might not have been representative of the general Indonesian population. In fact, the results were observed in a group predominantly composed of young, highly educated females belonging to upper–middle to high socioeconomic levels. However, comparable findings were reported by Fancourt et al. [117] for the beginning of the lockdown period. Furthermore, Fancourt et al. [117] observed a rapid decline in anxiety symptoms, likely because of the adaptation to the restrictions. Interestingly, Anindyajati et al. [73] observed that professionals working on the front line to combat the pandemic exhibited a lower risk of developing anxiety, perhaps due to the continuous exposure to information and updates related to the pandemic.

Overall, it appears that anxiety symptoms were greater during the first stages of the lockdown. Once the population became familiar with the restrictions imposed by the governments, the symptoms tended to decrease. Ultimately, higher anxiety levels were recorded when transitioning to the post-lockdown stage, where people were required to integrate features of their pre-COVID routines with the newly acquired needs (e.g., remote working and implementation of technologies for daily living).

##### Other Conditions

Frequently, when dealing with stressful situations, people may adopt maladaptive coping solutions, including engaging in self-harm behaviors. In line with this, a study conducted in England gathered data on hospital presentations associated with self-harming behaviors before, during, and after the initial lockdown [86]. The results highlight a reduction in access to health services, especially during the first weeks after the government restrictions were introduced. The authors discussed two possible explanations for this: on the one hand, the perception of a broader threat could have increased the sense of belonging and the concern for other people facing the same battle, which, in turn, acted as a protective factor, especially for the younger population. On the other hand, the reduction in hospital presentations could be a solution to avoid the risk of contagion. These results are in line with other evidence in the literature regarding hospitals in England (e.g., ref. [118]) and in other countries worldwide (e.g., refs. [119,120]).

Alcohol and drug misuse is also a maladaptive strategy for facing difficult situations. An investigation based on an Australian sample of people seeking treatment found an increased frequency of cannabis and illicit drug usage during the lockdown, [100]. In addition, alcohol was used to reduce depressive symptoms, loneliness, boredom, and anxiety, while low resilience corresponded to greater consumption of methamphetamine. Although the authors note a potential selection bias in their sample, given the small response rate, these findings are in line with other investigations in different countries, such as in Italy, where access to the emergency department was mainly due to alcohol and substance abuse compared to other severe mental illnesses, even during the long post-lockdown period [93].

#### 4.1.2. Parental and Caregivers’ Mental Health

The solutions implemented to curb the spread of COVID-19 put a strain on individuals responsible for the care of vulnerable or younger individuals due to increased caregiving demands, which, in the long run, can be exhausting. An investigation of Thai mothers with infants from 0 to 12 months calls attention to some situations that affected maternal mental health during and after the lockdown and which, in turn, could alter the psychological development of the baby [82]. For instance, difficulties in facing household expenses and bills, as well as household crowding during the lockdown period, resulted in worsened psychological conditions in mothers. This is especially true when considering that the sample of participants taking part in the study was not representative of the population of Thai mothers with poor socioeconomic levels and a lack of social support [82].

Another group of caregivers whose mental health deteriorates during and after periods of domestic isolation are those providing care for individuals with cognitive impairment and dementia. Bao et al. [90] observed diversified caregiver profiles according to the nature of the dementia being cared for. Caregivers of individuals with Lewy body dementia displayed the highest levels of distress and care burden compared to those caring for people with Alzheimer’s. This difference is likely due to the fact that Lewy body dementia often results in severe psychiatric issues and sleep disorders [90].

Another population at risk of aggravated mental health is that of “shielders”, referring to professionals working on the front line to control the contagion. During the second wave of COVID-19, shielders exhibited heightened distress and increased fear of contamination, leading to generalized anxiety due to exposure to the virus [75]. Moreover, Daniels and Rettie [75] found that anxiety in shielders tended to increase over time, with a reduced ability to find and adopt effective coping mechanisms.

#### 4.1.3. Resilience in Children and Adults

The study by Schnell and Krampe [63] conducted during the first lockdown in Germany and Austria focused on the concepts of “meaningfulness” and “self-control” as mediators of the association between acute stress caused by the pandemic and general mental health. Their results highlighted higher psychological distress after the lockdown compared to the in-home period, suggesting a persisting destabilization. Moreover, they noticed that higher rates of meaningfulness and self-control acted as protective factors. Conversely, the perception of low self-control when dealing with COVID-19-related stress could buffer existential strains with a subsequent development or worsening of anxiety and/or depression. Other protective factors identified in the literature are resilience, self-efficacy, hope, optimism, peaceful disengagement, and wisdom [89]. Conversely, Silveira et al. [9] highlighted the role played by social cohesion and adaptive coping strategies in stress buffering. People showing greater levels of social cohesion before the pandemic showed more negative effects on mental health during the domestic isolation period, while a higher score in social cohesion during and after the lockdown was correlated with more adaptive strategies for psychological recovery.

Overall, it appears that stressful and protracted situations can stimulate the emergence of individual resources. This process is referred to as “post-traumatic growth” and has been investigated in carers of children, mainly parents, during the first weeks following the ease of the restrictive measures. Among the positive outcomes, the most reported was an increased closeness within family relationships, followed by a reconsideration of personal priorities and values, and the possibility of adopting a healthier and more parsimonious lifestyle [74]. Regarding the support and promotion of resilience in children, the evidence suggests that parents with stronger personal competencies were better equipped to foster the development of better individual resources in their children [87].

#### 4.1.4. Difficulties in “Going Back to Normality”

The issues derived from the process of readjustment after the fear and isolation experienced during the peak of the pandemic were focused on by Dumitrache et al. [67]. In their study on the post-lockdown effects on the mental health of Romanian students, they observed, especially in females, an association between perceived stress and reduced social interactions, boredom, increased interest in following the pandemic’s broadcast, and longer times spent on phone calls. Moreover, a mediation analysis emphasized that the interest in the news related to COVID-19 modulated the link between boredom and stress. This picture highlights the situation 6 months after the end of the forced in-home isolation, suggesting that maladaptive coping mechanisms may arise from the reduced ability to organize an optimal response due to contextual negativity [67].

Prolonged domestic self-isolation has led, in some cases, to nocturnal post-traumatic stress disorder-like manifestations [81]. An exploration of sleep quality and oneiric activity found that the pandemic altered the content of dreams, often with fear as the main emotion [113]. A relatively stable persistence of depressive and anxious symptoms, as well as overall poor sleep quality, were observed in the post-traumatic aftermath of the pandemic [113].

#### 4.1.5. Defining and Profiling Guidelines for Mental Health Services

Undeniably, the pandemic event caused by the spread of COVID-19 had (and still has) severe consequences. Other than physical diseases, the occupational, economic, educational, and psychological fields have been critically placed under pressure. In regard to mental health, three major disorders, namely depression, anxiety, and stress, have affected (to a different extent) a large portion of the global population. As such, researchers and practitioners have collaborated with governments and national institutions to find solutions to face the present threat and define protocols to prevent post-pandemic emergencies and curb psychological disorders. For instance, Sasaki et al. [121] reported that, following the evidence based on pre-disaster preparedness adopted in Japan for earthquakes and tsunamis, investing in preventive instrumental and emotional support might be beneficial to mitigate the onset of mental disorders in the case of traumatic events. Moreover, Samy et al. [85] described a series of health protocols implemented in Asian countries to provide psychological support and establish a strategic plan for guaranteeing mental health services, even during extended domestic isolation or with reduced accessibility to hospitals. The need to find short-term solutions to buffer the critical pandemic situation has led to the adoption of hybrid interventions and the reinforcement of telemedicine, activating, for instance, 24/7 phone helplines [80,85].

The major citing documents in clusters no. 4, no. 11, no. 17, no. 18, no. 43, no. 47, and no. 51 are presented in Table 4. The following is a discussion of each cluster.

### 4.2. Cluster No. 4: Novelty Seeking and Support System

This cluster (LLR title = ”Resilience”; silhouette = 1.000), consisting of two citing articles, focuses on two different themes: novelty seeking and support system, each addressed by one of the two documents.

Li et al. [64] conducted a longitudinal study to examine changes in novelty seeking over three distinct time periods: before, during, and after the lockdown. Across all time points, higher levels of novelty seeking were associated with fewer depressive, anxious, and stress-related symptoms. Novelty-seeking reached its peak in the post-lockdown period. Stress, anxiety, and depression decreased during the lockdown and increased again in the post-lockdown period. Overall, the results seem to suggest that novelty-seeking acts as a protective personal factor against negative mental health outcomes. However, the causal direction of the relationship between the levels of novelty seeking and mental health cannot be determined from the cross-sectional design of the study [64].

Suhail et al. [88] showed that, in the Indian population, support from families, friends, and significant others, functioned as a protective factor in reducing the risk of developing psychiatric symptoms (e.g., somatic symptoms, anxiety, and depression) during the COVID-19 pandemic. Other studies in the literature have highlighted the importance of social support against negative mental health outcomes. For instance, Elmer et al. [122] observed that isolation in social networks, the lack of interaction and emotional support, and physical isolation were associated with negative mental health trajectories in a sample of Swiss undergraduate students.

**Table 4 ijerph-20-06310-t004:** Citing articles in other clusters identified using the DCA.

Cluster	Citing Article	Year	GCS	Coverage
*Cluster no. 4*				
	Li et al. [64]	2020	31	48
	Suhail et al. [88]	2022	6	1
*Cluster no. 11*				
	Chong et al. [123]	2020	30	34
*Cluster no. 17*				
	Presti et al. [124]	2020	68	28
	Cimino et al. [125]	2022	0	1
*Cluster no. 18*				
	Chen et al. [65]	2020	23	28
*Cluster no. 43*				
	Ezpeleta et al. [126]	2020	59	15
*Cluster no. 47*				
	Allan et al. [127]	2020	53	12
*Cluster no. 51*				
	Anyan et al. [128]	2020	59	15
	Carlyle et al. [100]	2021	4	1

### 4.3. Cluster No. 11: Pediatric Hospitalization

This cluster (LLR title = ”During”; silhouette = 1.000) consists of only one citing article authored by Chong et al. [123]. The authors focused on the admissions at pediatric hospital services due to the impact of COVID-19. Specifically, the investigation concerned access to the emergency department, diagnostic resources, and the use of hospital facilities in Singapore. The findings highlighted a considerable drop in admissions related to the virus contagion, which was most likely due to the lockdown, as well as an increase in the relative share of trauma-related attendances. A reduction in admissions at pediatric hospitals during lockdown was observed worldwide both in regards to infectious diseases as well as other conditions [129].

### 4.4. Cluster No. 17: Emotion Impact and Transmission

Cluster no. 17 (LLR title = ”Dynamics”, silhouette = 0.999), consisting of two citing papers, was renamed “Emotion Impact and Transmission”. The article by Presti et al. [124] discussed the dynamics of fear during the pandemic from a theoretical perspective. For instance, the authors noted that we react to fear symbolically and that we link emotions to other things and events using linguistic associations. Thus, language can modify the way we experience events and, as a result, affect how functionally or dysfunctionally oriented we are in the context that surrounds us.

Reference [125] explored the effects of COVID-19 (from before, during, and after the outbreak) on the psychological symptoms in school-age children and their mothers by using self-report instruments. The authors created two experimental groups: mothers at-risk for psychopathology and non-at-risk mothers. In doing so, the authors aimed to test whether the psychopathological risk of parents was associated with increases in behavioral problems among children during the pandemic. At-risk mothers exhibited a generally decreasing trajectory of psychopathological symptoms and related behaviors over time. However, there were no significant changes in children’s aggression or depression levels when comparing the three time-points. In the group with non-at-risk mothers, children’s aggression levels were lower after the quarantine period, while scores related to depressive symptoms were significantly higher during and after the lockdown. Taking these findings together, it appears that maternal risk for psychopathology does not affect specific aspects of children’s emotional and behavioral functioning; rather, it impacts the overall psychological well-being of their offspring.

### 4.5. Cluster No. 18: Mental Health Services

Cluster no. 18 (LLR title = mental health service; silhouette = 1.000), consists of one citing article [65]. The paper evaluated the medium-term influence of COVID-19 on admissions to mental health centers in secondary care. According to the results, after an initial decrease in mental health referrals, an acceleration in the referrals—meaning access and permanence in hospital wards—was observed. This acceleration was defined as the variation of daily referrals, computed as follows:$$acceleration\left(x_{1}\right)\;=\;\frac{new\_referrals\left(x_{1}\right)-new\_referrals\left(x_{0}\right)}{x_{1}-x_{0}}$$
where $x_{1}$ represents the day that we are interested in and $x_{0}$ is the day before. It is measured in $referrals/day^{2}$, where the term “new referrals” corresponds to new daily admissions and is used to distinguish new requests from previous ones. The acceleration was significant for both males and females, especially for individuals of working age, with Caucasian ethnicity, living alone, and with a pre-existing diagnosis of depression. There was no substantial post-lockdown acceleration detected in children, adolescents, older individuals, or ethnic minorities.

### 4.6. Cluster No. 43: Adolescents

This cluster (LLR title = ”Spanish Adolescent” silhouette = 1.000) consists of one citing article [126]. Ezpeleta et al. [126] investigated the mental health of adolescents following the period of in-home closure in a sample of Spanish adolescents. The results of the study report worsened mental health in terms of behavioral conduct, prosociality, and peer relationships. In addition, higher conduct problems (e.g., fights, bullying of other children, lying, cheating, tantrums) were linked to maladaptive relationships in adolescents (e.g., deteriorated connections with family members, failure when communicating with friends online), parental behavior at home (e.g., arguing, giving up enforcing rules), and adolescent activities (e.g., difficulties in keeping up with daily routines, excessive screen time exposure). However, the authors noted that the results might not be generalizable to adolescents with lower socioeconomic statuses [126].

### 4.7. Cluster No. 47: Healthcare Workers

Cluster no. 47 (LLR title = ”Healthcare Worker”, silhouette = 1.000), consists of one citing article [127]. Allan et al. [127] investigated the prevalence of psychological symptoms in healthcare workers servicing wards where COVID-19 patients were treated. The authors adopted a longitudinal study consisting of 3 time periods: 1.5–6.9 months after the pandemic’s peak, 6–11.9 months after the peak, and at least 12 months after the peak of infections. According to the results, the predominant symptoms included post-traumatic stress symptoms and general psychiatric caseness. Similarly, Rossi et al. [130] observed that almost half of the recruited healthcare workers reported post-traumatic stress symptoms. In the same study, almost a quarter of the sample reported symptoms of depression, while one-fifth reported symptoms of anxiety.

### 4.8. Cluster No. 51: Physical Activity and Substance Use

Cluster no. 51 (LLR title = ”Change”, silhouette = 0.998) consists of two citing articles and focuses on behavioral changes during the lockdown, especially physical activity and substance use. In particular, Anyan et al. [128] analyzed the correlation between changes in physical activity and the corresponding anxiety and depressive symptoms, as well as the levels of resilience found in response to the restrictions adopted during COVID-19. The authors found that lower levels of physical activity were associated with a higher risk for depression and anxiety symptoms, especially for younger participants. However, higher resilience was a protective factor against depression and anxiety symptoms. With a review of studies conducted during the first year of COVID-19, Marconcin et al. [131] found similar results, with higher physical activity being associated with higher well-being, quality of life, and lower depressive symptoms, anxiety, and stress, independent of age.

Reference [100] investigated the relationship between changes in seeking and using substances, such as alcohol, during the first COVID-19 closure, and related psychological symptoms, in terms of depression, anxiety, and resilience. In the post-pandemic phase, people reported higher use of tobacco and cannabis. Lower cannabis usage was associated with higher anxiety symptoms and lower resilience scores.

## 5. Limitations of the Study

It is essential to mention that the present work has some limitations. First, one limitation of the study is the Scopus document selection mode. To retrieve documents, we searched for specific key terms in publication titles, abstracts, and keywords. For this reason, the present analysis might have unintentionally excluded some relevant documents that were not captured by the adopted key terms in the search string. This is especially relevant considering the challenge of finding a homogeneous set of key terms to retrieve literature studies on mental health in the post-lockdown period. This challenge was amplified by the relatively recent nature of the post-lockdown period as well as the temporal diversities across countries, in terms of contagious waves and lockdown restrictions. For this reason, among the 791 documents retrieved, some dealt with the period following the first lockdown, others were related to a subsequent lockdown, and others were related to both.

Another limitation regards the choice of the platform used for conducting the literature search, which was Scopus, in our case. Some relevant documents might have been unintentionally excluded because they were published in journals or books that are not indexed in Scopus [132].

Furthermore, the temporal sequence of events did not make it possible to create a study with stable and well-defined data. Parameters, such as citation bursts, were temporally incomplete and, therefore, needed to be interpreted cautiously, as the network documents were very recent. From this point of view, most of them present a burstiness time frame that began in 2020 or 2021 and may not have ended yet. As such, it would be interesting to replicate the scientometric analysis in a few years, to assess the consideration by other scientific works in the longer term.

Lastly, while the DCA provides a quantitative analysis of co-citation frequencies, no information about the qualitative relationships among documents can be obtained from the software [27]. For this reason, scientometric experts advise integrating a qualitative discussion, as was conducted in this study, to better understand the content of the clusters [133].

## 6. Conclusions

Despite the novelty of the COVID-19 phenomenon, many research studies on the psychological and psychiatric consequences of the pandemic were published during the time frame considered in this study. The implications of the present study in the clinical setting can be extended to the analyzed symptom classes in the various clusters. The pandemic has widely affected well-being, and it would be relevant to continue monitoring the mental health of the population, including both clinical and healthy individuals. Furthermore, more research is needed to understand the impact of COVID-19 restrictions on populations with lower socioeconomic status. Looking closer at the articles that emerged from the scientometric analysis, it appears that, while the common thread is represented by mental health conditions related to the pandemic, the topics are diversified and include efforts to find more accessible treatment solutions and prevention strategies in case of further global challenges. As such, it appears that social connectedness acts as a common denominator when facing worldwide adversities, suggesting that both individual and collective factors should be taken into account. Accordingly, social connectedness played a protective role during COVID-19 lockdown restrictions by reducing negative physical and mental health outcomes [134]. On the organizational level, the pandemic emergency exposed the vulnerabilities of national health systems and the planning of international guidelines, leading to even more severe consequences [135]. This emergency presents an opportunity for future research to assess and improve healthcare systems [136,137]. Future studies should also focus on the developmental trajectories of children and adolescents to determine the long-term consequences of the pandemic and post-lockdown period on education, relationships, and mental health.

## Figures and Tables

**Figure 1 ijerph-20-06310-f001:**
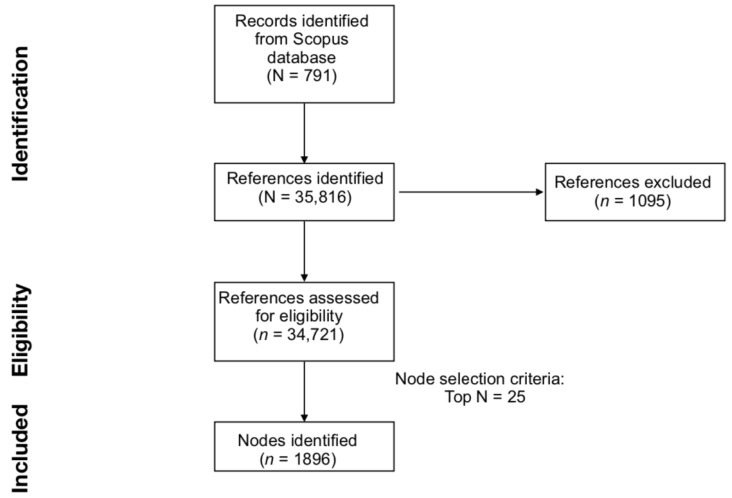
Graphical representation of the literature search, generation of the DCA network generation, and evaluation phases.

**Figure 2 ijerph-20-06310-f002:**
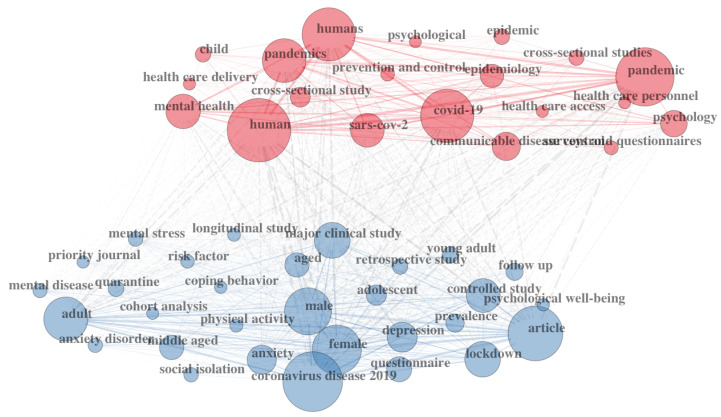
Graphic representation of the co-occurrence of keywords among citing documents.

**Figure 3 ijerph-20-06310-f003:**
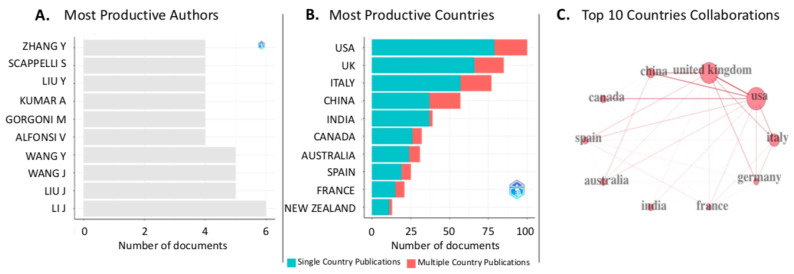
Graphical results of the bibliometric analysis conducted using the *bibliometrix* package for R [25]. (**A**) Most productive authors; (**B**) most productive countries; (**C**) collaborations among countries are limited to the top 10.

**Figure 4 ijerph-20-06310-f004:**
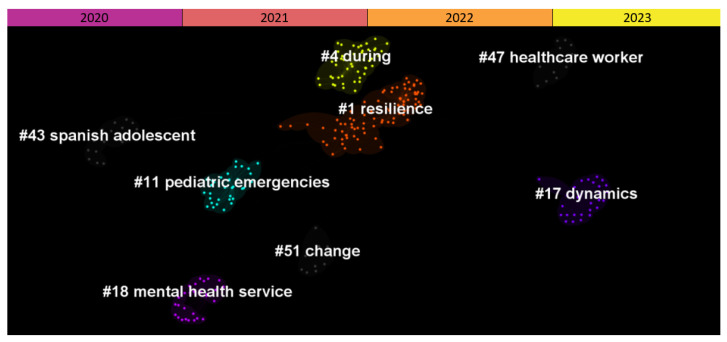
Document co-citation analysis network of the literature found in Scopus about mental health in the post-lockdown period. The network is divided into 8 thematic clusters of research.

**Table 1 ijerph-20-06310-t001:** Main statistics of the clusters identified within the document co-citation analysis network.

Cluster ID	Size	Silhouette	Mean Year	LLR Label	LLR Value	Suggested Label
1	86	0.853	2020	Resilience	18.00	Long-term Effects and Resilience
4	48	1.000	2020	During	16.68	Novelty Seeking and Support System
11	34	1.000	2020	Pediatric Emergencies	14.83	Pediatric Hospitalization
17	28	0.999	2020	Dynamics	24.09	Emotion Impact and Transmission
18	28	1.000	2020	Mental Health Service	11.38	Mental Health Services
43	15	1.000	2020	Spanish Adolescent	14.15	Adolescents
47	12	1.000	2020	Healthcare Worker	13.00	Healthcare Workers
51	11	0.998	2020	Change	24.55	Physical Activity and Substance Use

**Table 2 ijerph-20-06310-t002:** Top 20 documents ordered by the highest citation bursts.

n	Reference	Citation Burstiness	Publication Year	Burst Begin	Burst End	Duration
1	Brooks et al. [45]	29.18	2020	2021	2023	2
2	Xiong et al. [49]	15.99	2020	2021	2023	2
3	Holmes et al. [46]	8.43	2020	2021	2023	2
4	Wang et al. [48]	7.46	2020	2021	2023	2
5	Huang and Zhao [3]	7.46	2020	2021	2023	2
6	Pierce et al. [47]	7.30	2020	2021	2023	2
7	Vindegaard and Benros [50]	6.92	2020	2021	2023	2
8	Rajkumar [51]	6.40	2020	2021	2021	<1
9	Cao et al. [52]	6.06	2020	2021	2021	<1
10	Casagrande et al. [53]	5.85	2020	2021	2023	2
11	Pfefferbaum and North [54]	5.15	2020	2021	2023	2
12	Fiorillo and Gorwood [55]	4.08	2020	2021	2023	2
13	Torales et al. [56]	3.62	2020	2021	2021	<1
14	Wang et al. [57]	3.53	2020	2021	2021	<1
15	Odriozola-González et al. [58]	3.53	2020	2021	2021	<1
16	Salari et al. [59]	3.02	2020	2021	2021	<1
17	Galea et al. [60]	3.02	2020	2021	2021	<1
18	Rogers et al. [61]	3.02	2020	2021	2021	<1
19	Jiao et al. [62]	2.93	2020	2021	2021	<1
20	Stanton et al. [6]	2.93	2020	2021	2021	<1

**Table 3 ijerph-20-06310-t003:** The citing articles included in cluster no. 1 resulting from the DCA.

n	Citing Article	Year	GCS	Coverage
1	Schnell and Krampe [63]	2020	66	52
2	Li et al. [64]	2020	31	51
3	Chen et al. [65]	2020	23	31
4	Min et al. [66]	2021	2	8
5	Dumitrache et al. [67]	2021	6	7
6	Charbonnier et al. [68]	2021	7	7
7	Silveira et al. [9]	2022	9	6
8	Zapata-Ospina et al. [69]	2021	9	5
9	Morris et al. [70]	2022	0	5
10	Liu et al. [71]	2021	3	5
11	Leightley et al. [72]	2021	4	5
12	Anindyajati et al. [73]	2021	15	5
13	Stallard et al. [74]	2021	36	4
14	Daniels and Rettie [75]	2022	1	4
15	Wang et al. [76]	2021	3	4
16	Fioretti et al. [77]	2022	0	4
17	Dalkner et al. [78]	2021	4	4
18	Fancourt et al. [44]	2021	401	4
19	Siddi et al. [79]	2022	0	4
20	Khanna and Jones [80]	2021	2	4
21	Gorgoni et al. [81]	2021	7	4
22	Sirikul et al. [82]	2021	0	4
23	Campos et al. [83]	2021	6	4
24	Žuljević et al. [84]	2021	8	3
25	Samy et al. [85]	2021	1	3
26	Hawton et al. [86]	2021	31	3
27	Mariani Wigley et al. [87]	2021	2	3
28	Suhail et al. [88]	2022	6	3
29	Pellerin et al. [89]	2022	1	3
30	Bao et al. [90]	2022	0	3
31	Khan et al. [91]	2021	0	3
32	Costanza et al. [92]	2021	12	3
33	Brandizzi et al. [93]	2022	1	3
34	Brindisi et al. [94]	2022	2	3
35	Caballero-Apaza et al. [95]	2021	2	3
36	Liu et al. [96]	2021	46	3
37	Gavin et al. [97]	2021	1	3
38	Molnar et al. [98]	2021	2	3
39	Escudero-Castillo et al. [99]	2021	36	3
40	Carlyle et al. [100]	2021	4	3
41	Cui and Han [101]	2022	0	3
42	Wang et al. [102]	2021	4	3
43	Xu et al. [103]	2021	5	3
44	Iodice et al. [104]	2021	23	2
45	Leonangeli et al. [105]	2022	1	2
46	Hildersley et al. [106]	2022	1	2
47	Dodge et al. [107]	2021	6	2
48	Chatterji et al. [108]	2021	6	2
49	Passarelli et al. [109]	2022	0	2
50	Yang et al. [110]	2022	0	2
51	Sempere et al. [111]	2021	1	2
52	Corley et al. [112]	2021	78	2
53	Gorgoni et al. [113]	2022	2	2
54	Mariani et al. [114]	2021	8	2
55	Dewa et al. [115]	2021	20	2

## Data Availability

Not applicable.

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
