# Peer review of "Mental Health in the Post-Lockdown Scenario: A Scientometric Investigation of the Main Thematic Trends of Research"

_ijerph, 2023, doi:10.3390/ijerph20136310_

Round 1
Reviewer 1 Report
To authors,
This study is a review study that visualizes COVID-19-related research from the last three years with bibliometric analysis.
The concept of eight clusters calculated by DCA is very appropriate and appears to be a comprehensive and good description of the mental health aspects of the COVID-19 pandemic.
1. Clarify the title of your paper to include your research question.
2. Please describe how the DCA analysis applied in this study is more meaningful than meta-analysis or other review methods.
3. Move the description of the eight thematic clusters currently presented in the Discussion section to the Results section, and include any additional comments from researchers in the Discussion section.
4. Is there anything special about your findings compared to other papers already published?
5. Describe any limitations of the study that arose during the data extraction process.
6. Enter the log-likelihood ratio for each label in Table 1.
7. When creating keywords, consider including terms such as bibliometrics, visualized research, or document co-citation analysis.
My comments can be taken as suggestions.
When creating keywords, consider including terms such as bibliometrics, visualized research, or document co-citation analysis.
I wish you all the best in your revision work and hope to see this manuscript as a published research article.
Sincerely, your reviewer
Reviewer 2 Report
In this interesting paper, the authors review the effect in the post-lockdown periods; we analyzed the available scientific literature using a scientometric and bibliometric approach. As they consider scientometry allows assessing the published research in a specific field, reducing the risk of biases and adopting a systematic and objective computational procedure. I was not familiar with this type of review and really I do not feel secure about statistics and computational procedures used. However its results are clear to me. In my opinion the paper can be published at it is, but only needs to improve all figures resolution.
This topic is relevant in the sense that the methodology used permits to cluster the subject/topic of publications and classified them in cluster of themes or affected populations and afterwards authors can qualitatively summarize the main results of these themes or populations and detect areas where more investigations are needed.
As far I have understood, what this new methodology adds to the subject is that the methodology permits to be more exhaustive to detect publications on an area of knowledge and summarize information by topic very easily.
Conclusions are consistent with the evidence and arguments presented. They adress some of the needs that are not covered by the literature reviewed.
There are some errors in the references as Abbreviated Journal Name (most are complete names) and some are not well written. Please review them.
Figures need to have higher resolution
Reviewer 3 Report
Dear Authors,
I find your manuscript rather interesting. I do have some comments and suggestions:
Introduction
I would suggest that this section could be extended. It is recommended that authors address public health issues related to specific mental disorders.
It seems necessary for authors to set out the research questions (RQ).
Results
The resolution in figures 2 and 3 should be optimized.
The results section describes the combination of inconsistent information. I would suggest that a distinction should be made between where the methods and results have been submitted.
Discussion
The text in this section was very difficult to follow.
All in all, please rewrite and organize your methods, results, and discussion sections.
Kind Regards
Round 2
Reviewer 3 Report
The authors did not fully answer my questions.
Although the investigative methods used are certainly interesting, the manuscript still has shortcomings.
The main comments are listed below
Results
Study characteristics: Cite each included study and present its characteristics.
Risk of bias in studies: Present assessments of risk of bias for each included study.
Certainty of evidence: Present assessments of certainty (or confidence) in the body of evidence for each outcome assessed.
Discussion
Provide a general interpretation of the results in the context of other evidence.
Discuss any limitations of the evidence included in the review.
Discuss any limitations of the review processes used.
Discuss implications of the results for practice, policy, and future research.
Best Regards
